# Dietary Isothiocyanates, Sulforaphane and 2-Phenethyl Isothiocyanate, Effectively Impair *Vibrio cholerae* Virulence

**DOI:** 10.3390/ijms221910187

**Published:** 2021-09-22

**Authors:** Klaudyna Krause, Agnieszka Pyrczak-Felczykowska, Monika Karczewska, Magdalena Narajczyk, Anna Herman-Antosiewicz, Agnieszka Szalewska-Pałasz, Dariusz Nowicki

**Affiliations:** 1Department of Bacterial Molecular Genetics, Faculty of Biology, University of Gdansk, 80-308 Gdansk, Poland; klaudyna.krause@phdstud.ug.edu.pl (K.K.); monika.szalkowska@phdstud.ug.edu.pl (M.K.); agnieszka.szalewska-palasz@ug.edu.pl (A.S.-P.); 2Department of Physiology, Medical University of Gdansk, 80-211 Gdansk, Poland; agnieszka.pyrczak-felczykowska@gumed.edu.pl; 3Department of Electron Microscopy, Faculty of Biology, University of Gdansk, 80-308 Gdansk, Poland; magdalena.narajczyk@biol.ug.edu.pl; 4Department of Medical Biology and Genetics, Faculty of Biology, University of Gdansk, 80-308 Gdansk, Poland; anna.herman-antosiewicz@ug.edu.pl

**Keywords:** *Vibrio cholerae*, cholera, sulforaphane, phenethyl isothiocyanate, (p)ppGpp, antimicrobial, mode of action, virulence, biofilm

## Abstract

*Vibrio cholerae* represents a constant threat to public health, causing widespread infections, especially in developing countries with a significant number of fatalities and serious complications every year. The standard treatment by oral rehydration does not eliminate the source of infection, while increasing antibiotic resistance among pathogenic *V. cholerae* strains makes the therapy difficult. Thus, we assessed the antibacterial potential of plant-derived phytoncides, isothiocyanates (ITC), against *V. cholerae* O365 strain. Sulforaphane (SFN) and 2-phenethyl isothiocyanate (PEITC) ability to inhibit bacterial growth was assessed. Minimum inhibitory concentration (MIC) and minimum bactericidal concentration (MBC) values indicate that these compounds possess antibacterial activity and are also effective against cells growing in a biofilm. Tested ITC caused accumulation of stringent response alarmone, ppGpp, which indicates induction of the global stress response. It was accompanied by bacterial cytoplasm shrinkage, the inhibition of the DNA, and RNA synthesis as well as downregulation of the expression of virulence factors. Most importantly, ITC reduced the toxicity of *V. cholerae* in the in vitro assays (against Vero and HeLa cells) and in vivo, using *Galleria mellonella* larvae as an infection model. In conclusion, our data indicate that ITCs might be considered promising antibacterial agents in *V. cholerae* infections.

## 1. Introduction

Cholera infection caused by the Gram-negative, aquatic bacterium *Vibrio cholerae* leads to severe dehydrating diarrhea [1]. The worldwide spread of *V. cholerae* has caused seven large pandemic events in recorded history and remains a serious threat to public health in developing countries [2,3]. Approximately 2 to 9 million cases of cholera occur annually, resulting in near 100,000 deaths [4]. The case-fatality ratio is correlated mainly with proper and rapid treatment, ranging from 15.8% to less than 1%, according to WHO statistics. Currently, the antibiotic resistance among *V. cholerae* strains is increasingly reported and threatens our ability to treat infected patients—which has created a major challenge to the development of an efficient therapy [5,6,7]. Oral rehydration treatment (ORT) is still the first-choice therapy for infected patients [8]. In more serious cases, intravenous rehydration and antibiotics are employed to reduce severe complication development. However, the ORT helps only to reduce symptoms of infection instead of actual treatment and cannot eliminate the cause of dysentery, while the continuous use of antibiotics contributes to a rapid emergence of antibiotic-resistant strains.

Cholera toxin (CT) and the toxin coregulated pilus (TCP) are two major virulence factors responsible for toxigenic abilities and intestinal colonization, respectively. Toxigenic abilities of *V. cholerae* are a consequence of CT synthesis encoded by the *ctxAB* gene, located on the CTX prophage integrated on the main chromosome. This has prompted many researchers to investigate the therapeutic potentials of traditional foods and beverages as well as natural products to inhibit the growth of *V. cholerae* or secretion of CT [5,9]. Moreover, virulence factors are regulated at the transcriptional level by a coordinated network of proteins that respond to specific environmental signals. Among these proteins is ToxT, an AraC family regulator that binds the promoters *tcpA* and *ctx* and directly activates virulence gene expression [1]. ToxT plays an essential role in *V. cholerae* virulence, and its expression and activity are highly regulated among epidemic *V. cholerae* strains.

*V. cholerae* strains are a common bacterial flora found in their natural reservoirs, both in freshwater and marine environments [10]. In these conditions, *V. cholerae* is mainly attached to the biotic surfaces, including those of aquatic species such as plankton, clams, crustaceans, plants, and insects [10,11]. The efficient propagation and resistance of bacterial cells in a nutritionally poor aquatic environment is facilitated by biofilm formation. It is assumed that the bacteria existing in the biofilm show a reduced sensitivity to environmental stresses, including digestion in the gastrointestinal tract, which is a route of infection; thus, the ability to form and maintain a biofilm is of serious clinical importance. Growth in biofilm induces a hyper-infectious phenotype in *V. cholerae* [12]; hence this feature and generally the higher number of bacteria in biofilm play a key role in transmission [13,14]. The hyperinfective state refers to a decrease in the number of cells required to cause disease. Thus, biofilm prevention and eradication are crucial steps in an anti-cholerae strategy. In an ongoing challenging era where microbial resistance to antibiotics is a significant problem in controlling infections caused by bacteria and in the occurrence of recurrent epidemics due to drug-resistant microorganisms, the development of new therapeutic strategies is of high priority. Targeting the regulation of bacterial virulence is considered one of the promising approaches [14,15].

Current antimicrobials in their mode of action target a relatively small number of essential cell functions such as DNA/RNA, cell wall, or protein synthesis. So, if these global process inhibitors are no longer effective, extensive efforts should be undertaken to investigate new and more specific targets. Phytoncides are a wide group of plant secondary metabolites which are a source of compounds with biological activities, including antimicrobial action. Among them, isothiocyanates (ITC), produced by Brassicaceae plants, have been shown to be effective against pathogenic bacteria and fungi [16,17]. Sulforaphane (SFN) is an aliphatic ITC available in the daily diet and abundant in the seeds and sprouts of cruciferous plants, such as broccoli [18]. 2-Phenethyl isothiocyanate (PEITC) is an aromatic member of ITC compounds, naturally occurring in watercress [19]. ITC compounds demonstrate numerous health benefits for humans, including anticancer, anti-inflammatory, and antioxidant properties [20]. Moreover, SFN and PEITC exhibit antimicrobial potential across Gram– and Gram+ bacterial pathogens like *Enterococcus faecalis*, *Staphylococcus aureus*, *Campylobacter jejuni*, or enteropathogenic *Escherichia coli* [16,21]. The mechanism of antibacterial action might vary in different bacterial species and includes cell membrane disruption, induction of general stress, and changing metabolism through stringent control activation [22,23,24]. In fact, it was demonstrated by us that the ITC-induced accumulation of the stringent response alarmones, ppGpp and pppGpp, underlies the antibacterial effect of ITC in bacterial pathogens [21,22,25]. The need for safe and effective compounds as therapeutic agents against drug-resistant microorganisms has driven many studies toward medicinal plants to date. The aim of this study was to comprehensively elucidate the antimicrobial effect of the food-grade natural compounds, SFN and PEITC, against the classical O1 biotype of *V. cholerae* O395 with a particular focus on their impact on virulence factors expression and biofilm control. Furthermore, we proposed an insight into their mode of action through activation of the bacterial stress response system—the stringent response. The effectiveness of SFN as a therapeutic agent was also examined in the *G. mellonella* infection model.

## 2. Results and Discussion

### 2.1. Sulforaphane and 2-Phenethyl Isothiocyanate Impair the Growth of V. cholerae in a Dose-Dependent Manner

The number of newly discovered or designed antimicrobials in recent years is dramatically low and even decreasing further. Thus, it can be expected that the imbalance between rising antimicrobial resistance among bacterial pathogens and the availability of treatment strategies will expand continuously [26,27]. Therefore, in this study, we decided to employ two members of ITC group, naturally occurring in *Brassicaceae* plants, SFN and PEITC, to test their antibacterial potency against *V. cholerae.* So far, their antimicrobial effect was demonstrated for such bacterial pathogens as *E. coli*, *E. faecalis, S. aureus, Salmonella Montevideo*, *and C. jejuni* [21,24]. Hence, we assessed the antimicrobial activity of ITC by microdilution sensitivity assay to obtain the inhibitory concentration values for the tested compounds. As shown in Figure 1a, the minimal inhibitory concentrations (MIC) of SFN and PEITC were 0.5 mM and 2 mM, respectively, while their bactericidal potency was disclosed at 4 mM and 10 mM (SFN and PEITC, respectively). Chloramphenicol was used as a positive growth inhibition control in this study, and its MIC was assessed as 4 µg/mL and the MBC was 8 µg/mL. These results were confirmed by the disc diffusion assay where SFN and PEITC were used in the concentrations of 30 µg/disk resulting in a 23 ± 2 and 13 ± 1 mm diameter zone of inhibition. This is consistent with results observed for chloramphenicol (30 µg/disk) expressed as 21 ± 2 mm diameter zone of inhibition. Next, we tested the assessed MIC and their fractions (1/2 to 1/16 MIC) to elucidate whether their action is dose-dependent. For this, the fresh cultures (optical density, OD_600_ = 0.1) were treated with SFN or PEITC in a range of concentrations, and the growth was monitored spectrophotometrically during 3 h and compared to untreated control (Figure 1b,d). This experiment showed that the statistically significant growth impairment was observed for concentrations even lower than MIC, namely ½ and ¼ MIC (for SFN and PEITC, respectively) effectively inhibited the growth of bacterial cultures in their log-phase. The antimicrobial action of ITC differs among bacterial species tested to date [21,24] and still remains poorly understood in terms of the molecular targets which decide about nature of their bacteriostatic and bactericidal action. However, as it has been previously reported, the bactericidal activity of aromatic glucosinolate-derived compounds, such as 2-phenylethyl and benzyl isothiocyanates, was higher in comparison with aliphatic glucosinolate derivatives, such as SFN, allyl- 3-butenyl, and 4-pentenyl isothiocyanates [28].

Similarly, the bactericidal effect of isothiocyanates derived from aromatic glucosinolates was more pronounced than the activity of those derived from aliphatic precursors [25,29]. One of the proposed explanations of this phenomenon is the activation of oxidative stress response and subsequent changes in cellular redox homeostasis. It might be caused by interaction between isothiocyanates and thiol groups of microbial proteins engaged in redox homeostasis, such as glutathione [30]. Moreover, it was proposed that aromatic glucosinolate-derived isothiocyanates had higher reactivity through their ability to act as electron donors from a benzene ring [31,32]. However, our examination of the level of reactive oxygen species (ROS) in *V. cholerae* treated with SFN or PEITC in the 2,7-dichlorofluorescein diacetate (DCFH-DA) assay did not show a significant elevation of oxidative stress (data not shown), which is in agreement with our previous analyzes performed using the *E. coli* strain [25]. As the ROS level correlates with the mode of action of antimicrobials; therefore, the death curve upon ITC action was further analyzed (Figure 1c,e) to determine bacteriostatic/bactericidal activity against *V. cholerae*. The ITC was added to fresh culture adjusted to ~10^7^ c.f.u/mL, then samples were collected, and viable cells were enumerated at indicated time points up to 24 h. The results of this assay led us to the conclusion that antimicrobial action of both ITCs (in a range of concentrations from MIC to 5 × MIC) is bacteriostatic as observed c.f.u. could not drop below the starting culture yield by 99.9% (≥3 log_10_), which is an arbitrary threshold for bactericides. Just a relatively high concentration of PEITC (10 × MIC) showed the bactericidal effect, but it could be difficult to consider it for in vivo use due to its cytotoxic effect. However, to date, bacteriostatic agents such as doxycycline, macrolides, or azithromycin are in routine use and recommended in cholera infection treatment (WHO).

The biofilm formation of *V. cholerae* plays a very important role in the survival of bacteria and their resistance to antimicrobials, as well as virulence level; thus, controlling biofilm by antimicrobial compounds is of crucial importance to prevent and combat infections. Therefore, the ability of ITC to affect biofilm was assessed in this study. Biofilms already formed by *V. cholerae* were exposed to the 1 or 5 × MIC of SFN or PEITC and the biofilm mass was assessed by crystal violet staining (Figure 1f). Both tested ITCs reduced the biofilm significantly, and the effectiveness of this action was already exhibited for 1 × MIC. In line with this observation, the viability of the biofilm-forming bacteria was also considerably reduced, as shown in the MTT assay (Figure 1g). Importantly, these effects were similar for PEITC and SFN. The ITCs were also tested for their ability to prevent biofilm formation. For this, compounds were added to the bacteria before the biofilm could be formed. This experiment showed that ITC can also reduce the transition of bacteria from sessile to biofilm state (Appendix A). SFN exhibited a stronger effect in this assay; however, PEITC at concentrations of 1 and 5 × MIC also affected biofilm formation. To date, the antibiofilm activity of ITC has been scarcely described in the literature, with some exceptions [33,34,35]; therefore, the molecular basis of its nature is poorly understood. Kaiser et al. [23] showed that PEITC is effective in inhibiting *P. aeruginosa* biofilm formation even in concentrations significantly lower than the MIC (≤29 mg/mL). The authors indicated a good ability of the tested compound to penetrate the complexed biofilm matrix. In another study [36] iberin, ITC from horseradish, was proposed as a quorum sensing (QS) modulator that specifically blocked expression of QS-regulated genes in *P. aeruginosa*. These observations indicate that ITC can be used not only to inhibit bacterial exponential growth but, more importantly, also to diminish their biofilm formation and eradicate already formed biofilms, which highlights the important preventive role of ITC against pathologies induced by *V. cholerae.* Moreover, except for water contamination, seafood is most commonly associated with cholera [37]. ITCs are considered effective natural food preservatives. Allyl isothiocyanate in vapor and liquid forms has demonstrated high bactericidal activity against various food spoilage microorganisms [38]. Among others, its effectiveness in the preservation of fish products was shown. In this light, our findings could be useful in limiting the risk of cholera as a foodborne illness.

### 2.2. ITC Affects the Cell Morphology of V. cholerae

The observed growth inhibition and bacteriostatic action during ITC treatment were further analyzed by TEM observation to reveal their impact on the bacterial surface and morphological changes. It has been reported by others that ITC antimicrobial action can affect the function of the bacterial membranes and lead to their increased permeability or leakages (for review [24]). We showed that SFN and PEITC treatment (at their MICs) of *V. cholerae* led to cell shrinkage and cytoplasm condensation as indicated in photographs (Figure 2). Moreover, 120 min of ITC treatment resulted in cytosol leakage on distal parts of cells. However, the extensive damage of membranes or the cell wall was not observed. Thus, we can assume that the cell envelope is not a primary target of the tested compounds. Saleh et al. [39] reported that ITC-rich *Moringa peregrina* seed aqueous extract in a time-dependent manner affected the morphology of *Salmonella enterica* with progressive structural disorganization of the cytoplasm and the appearance of wavy membranes. In another study, Borges et al. [31] showed that aliphatic allyl-ITC (AITC) and PEITC were able to affect *E. coli* CECT 434 cell membrane integrity in a dose-dependent manner. The authors assumed that AITC and PEITC increased the hydrophilic character of the membrane, changing its physicochemical characteristics. However, it cannot be excluded that prolonged ITC treatment impacts cell morphology which is a secondary effect of these antimicrobials.

### 2.3. Starvation as a Mode of ITC Action against V. cholerae

It has been proposed that cytosol shrinkage might be one of the results of cell starvation [40]. According to Shi and colleagues, the condensation of cytoplasm is a natural feature of cells experiencing nutrient depletion. However, the prolonged metabolic stress observed in starved cells might lead to membranes dysfunction and loss of their potential [41,42]. In our previous studies, we showed that ITC treatment resulted in stringent response activation and a drastic decrease in nucleic acids synthesis as a consequence of the accumulation of starvation alarmone, (p)ppGpp [22,25]. Thus, we compared SFN and PEITC effect on RNA/DNA synthesis in [^3^H] radiolabeled nucleotide assay. A significant reduction of stable RNA synthesis was observed (Figure 3a,b), which indicates that general downregulation of major metabolic processes occurs rapidly after SFN or PEITC treatment as a result of the global stress response. We also observed the impaired DNA synthesis in treated cultures as assessed by [^3^H] thymidine incorporation (Figure 3c,d). To determine whether the stringent response alarmone accumulated after the ITC treatment, we measured the levels of the [^32^P] labeled nucleotides in the treated cells. This examination showed that under treatment with SFN or PETC, the (p)ppGpp alarmone levels increased and were comparable to those observed in *V. cholerae* cultures experiencing amino acid deprivation, which was induced by the addition of serine hydroxamate (SHX), the toxic analog of L-serine. Namely, the SFN or PEITC treatment for 15 min (at MIC) resulted in the accumulation of ppGpp (±162.5% and ±54.5%, respectively, in relation to cells treated with SHX). What is noteworthy is that both ITCs at concentrations lower than their MICs were able to significantly elevate the alarmone synthesis. SFN at 1/2, 1/4, and 1/8 of MIC elevated ppGpp level up to 168.5%, 116.7%, and 76.3%, respectively (Figure 3e). The cells treated with PEITC at the 1/2, 1/4, and 1/8 of MIC showed different dynamics of alarmone accumulation, namely 72%, 159.7%, and 117.6%, respectively (Figure 3f). The amounts of the detected guanosine nucleotides correlated with the inhibition of nucleotides synthesis in PEITC treated cells. At higher doses of PEITC (1/2 or 1 MIC), the GTP/GDP spots visible on chromatograms gave apparently lower signals, which suggests that the process of their synthesis was disturbed (Figure 3f).

The activation of the stringent response results in modulation of gene expression and dramatically changes metabolic pathways by recruitment of alternative sigma factors [43]. Therefore, the observed drop in DNA synthesis in our study might be connected to the indirect effect of the inhibition of gene expression and alarmone accumulation. (p)ppGpp generally acts as a signal of nutritional deprivation, but also physical, and other environmental changes can induce its synthesis. Its levels increase upon nutrient downshifts and are inversely correlated with growth rate [43,44]. In *V. cholerae*, (p)ppGpp is synthesized by two enzymes conservative for enterobacteria, RelA and SpoT, and the third one is RelV, which to date seems to be specific just for this species. In the recent studies, we showed that ITC action in *E. coli* relies on amino acid starvation pathway mediated by RelA [22,25], and *relA* deficient strain was unable to accumulate (p)ppGpp in response to ITC. It indicated that the stringent response induction by ITC is mediated by amino acid starvation. Antimicrobial activity of SFN and PEITC was also affected by the supplementation of a culture medium with several amino acids and tri-peptides [22,25,45]. Isothiocyanate group (-NCS) was proposed to be an electrophilic functional group prone to react with both amines and thiols [46,47]. Nonetheless, the interactions with specific cellular targets were also reported [48,49]. Luciano and colleagues showed direct enzymatic inhibition of thioredoxin reductase and acetate kinase in enterohemorrhagic *E. coli* O157:H7 by AITC [49]. We can assume that the observed morphological changes and metabolic downregulation of nucleic acid synthesis in *V. cholerae* as the response to tested ITC are connected with rapid elevation of (p)ppGpp levels. However, the genetic and molecular background of this phenomenon should be further investigated, especially the role of *relV* in the stringent response mechanism of *V. cholerae*.

### 2.4. The SFN and PEITC Treatment Controls Expression of Virulence Factors of V. cholerae

Rehydration and antibiotic therapy remain the first line of treatment of cholera infections. This strategy is targeted to cure the symptoms but remains blind to the serious problem, namely, the toxigenicity of bacteria due to CT production, which, in consequence, leads to severe health complications after the therapy and slows the patient convalescence. To gain insight into the potency of ITC to reduce the pathogenicity of *V. cholera,* we performed a quantitative RT-qPCR analysis of virulence genes expression (Figure 4a). The virulence of *V. cholerae* is maintained by several factors where the ToxT, ToxR, and ToxS regulatory proteins remain crucial for *ctx*AB genes expression called then the “ToxR regulon” after the first identified positive regulator [50].

Thus, the following genes were chosen according to their role in *V. cholerae* pathogenicity: *toxT, toxR, toxS*, encoding regulatory proteins; *ctxB* the CT beta subunit; *tcpH*, *tcpP,* and *tcpA* as involved in toxin coregulated pilus synthesis. Moreover, because of described significant role of (p)ppGpp alarmone in effective transcription from ToxR regulon, we decided to study the genes expression pattern also in amino acid starved cells during SHX treatment. Our results indicated that a high level of (p)ppGpp, elevated due to starvation signal from RelA activity, resulted in a general decrease in the expression of tested genes in SHX-treated cells. This downregulation was observed in SFN and PEITC challenged cultures (Figure 4a). The strong and statistically significant downregulation of *toxT*, the main toxin synthesis regulator, was noticed in a relatively short time, 15 min after the addition of compounds, namely −3.1, −7.6, and −25 fold change for SFN, PEITC, and SHX, respectively. The prolonged treatment (60 min) resulted in a less pronounced effect, especially for PEITC and SHX. Expression of *toxS* and *toxR* was downregulated efficiently by SHX and SFN, while PEITC reduced the expression of the *toxS* gene only. Notably and importantly, for the prospective treatment of cholera with ITC, the toxin gene expression was reduced significantly by PEITC and SHX already 15 min after addition of compounds, while for SFN, the effective downregulation of *ctxAB* gene expression required longer (60 min) exposure. The expression of genes involved in the synthesis of coregulated pilus was inhibited most effectively by SHX and SFN after longer treatment and by PEITC earlier. The Western blot analysis was used to confirm the downregulation of CT synthesis. Figure 4b shows that treatment of *V. cholerae* with SFN or PEITC resulted in significant impairment of toxin synthesis after 20 h (80% decrease compared to the control). Comparable results were observed for cells treated with SHX, where CT production decreased by 60%. In conclusion, expression of the genes involved in *V. cholerae* virulence is downregulated by all tested compounds–ITC and amino acid starvation is induced by SHX. This is expected to reduce the pathogenicity of the bacteria.

It was suggested by others that a high basal level of cellular (p)ppGpp positively correlates with toxin synthesis and *V. cholerae* virulency [51]. Nevertheless, even *relA* deficiency was not sufficient to reduce *V. cholerae* motility, biofilm formation abilities and pathogenicity in the murine model. This apparent discrepancy can be, however, explained as Fernández-Coll and Cashel [52] described distinguished differences in the effect of alarmone accumulation on gene expression when it happens as a natural process responding to environmental nutrient availability and growth rate, and in “sudden burst” scenario in response to harsh stress conditions. Thus, we could conclude that ITC treatment leads not only to growth inhibition but also affects the virulence genes expression through the rapid elevation of cellular (p)ppGpp level.

### 2.5. ITC Reduces the Toxicity of V. cholera In Vitro and In Vivo

Cholera infection symptoms strictly correspond to effective CT production in the host organism. Thus, downregulation of toxin synthesis together with growth inhibition by antibiotic compounds could provide higher efficiency than the traditional treatment. To evaluate the ITC effectiveness as anti-cholera agents, we checked their activity in human and simian cell lines, HeLa and Vero. These cell lines are suitable to test *V. cholerae* pathogenicity because of their sensitivity to CT, and they are also routinely used to test the cytotoxicity of different compounds [53,54]. We showed a significant reduction of the toxic effect of bacterial lysates treated prior with any of the compounds, SFN and PEITC (Figure 5). Observed results were similar in both cell lines. The slight differences in the activity of SFN and PEITC between cell lines could be related to the variation of the glycolipid receptor (GM1) expression on the cell surface, which may impact their sensitivity to CT [55,56]. These findings are in line with results showing toxin expression impairment presented above. Moreover, we show that ITC is not toxic by itself for eukaryotic cell lines in doses employed in this study.

Further, in order to determine if the inhibition of virulence assessed in vitro could also affect in vivo infection, we tested the ability of SFN to inhibit virulence of *V. cholerae* in a surrogate host of *Galleria mellonella* larvae. This model organism was described as a suitable alternative for vertebrates in selecting new antimicrobials as well as for *V. cholerae* infections approaches [57,58]. We first established the bacterial load appropriate for survival experiments (Appendix A). The 5 × 10^2^ bacteria per larvae were chosen for further analyzes as they resulted in a 50% death rate at 24 h after infection. SFN safety (at the 25, 10, and 5 mg/kg doses) was assessed, and the tested concentrations were found to be non-toxic, and no symptoms of melanization–associated with the worsening of the insect condition were observed after the treatment (data not shown). This dose corresponds with results obtained in the rodent model, assessed by others [59,60]. We demonstrate that the pretreatment of larvae with 25 mg/kg SFN administrated 3 h before infection improved the survival of *G. mellonella* larvae by 60% after 48 h and by 30% after 72 h post infection (Figure 6). Additionally, we analyzed the potency of SFN to enhance azithromycin (20 mg/kg) activity against *V. cholerae*. Azithromycin is considered an antibiotic of choice for diarrheal treatment in children [61] and is now under clinical trial investigation. The observed larvae survival rate after a single dose of azithromycin treatment was surprisingly similar to the level observed in the objects which received SFN alone. Interestingly, administration of two compounds together decreased mortality in *V. cholerae* challenged larvae to 20% after 96 h as compared with infected control.

The SFN is known for its anti-inflammatory and antioxidant properties but is also described as an immune response stimulator [62]. Ali et al. [63] showed improved clearance of *E. coli* and *S. aureus*, where SFN treatment significantly reduced intracellular bacteria survival of THP-1 and PBMC-derived macrophages. Our results show significantly enhanced hemocytes infiltration to hemolymph 3 h after SFN treatment (Figure 6c) compared to PBS treated control larvae. We also observed the 27.7% enhancement of phagocytosis rate (Figure 6d,e) by hemocytes obtained from *G. mellonella* injected with SFN. These results were assessed by fluorescence microscopy analysis of heat-killed FITC-labeled *V. cholerae* bacteria. But yet, the detailed mechanism of this observation should be elucidated to define SFN-triggered cellular factors. Deramaudt et al. [64] proposed a combinatory mechanism where reduction of the inflammatory response through modulation of p38 and JNK signaling pathways accompanied antibacterial action of SFN in mice macrophages infected with *S. aureus*. A similar mode of action may operate in *G. mellonella,* especially as this organism possesses counterpart MAP kinases, and they are activated in response to infection with the entomopathogenic bacterium *Bacillus thuringiensis* [65,66] Finally, the dosages chosen for in vivo virulence assay were comparable to those used and reported as safe to treat human diseases. Therefore, based on its effects, as well as a low toxicity, SFN could be considered as a support to current therapies for the management of cholera as monotherapy and for the combination with other antibiotics. However, further studies are needed to elucidate the molecular mechanisms underlying the activity of ITCs and also to identify their new molecular targets.

## 3. Materials and Methods

### 3.1. Bacterial Isolates and Growth Conditions

*V. cholerae* O395 (WT) [67] and *toxT*::*lacZ* [68] mutant were grown in LB medium (Sigma Aldrich, USA) at 37 °C or 30 °C (as indicated below) with aeration and shaking or plated on LB agar (supplemented with 1.0% bacteriological agar) and incubated overnight at 37 °C or 30 °C. For assessing the (p)ppGpp accumulation MOPS a medium with 0.2% casamino acids was used. For determination of susceptibility tests Mueller–Hinton (Sigma Aldrich, Burlington, MA, USA) medium was used.

### 3.2. Determination of Bacterial Growth Inhibition

The mechanism of ITC antimicrobial activities was tested according to CLSI standard methodology M07–A10. The MIC (minimal inhibitory concentration) was assessed by the twofold broth microdilution methods as described previously [69]. Mueller medium was used to prepare bacterial inoculums and to dilute the PEITC and SFN stock solution. Briefly, fresh *V. cholerae* culture containing 0.5 McFarland of bacteria was diluted 200 times, the Mueller–Hinton medium was added in equal volume to prepared twofold dilution of SFN and PEITC. Initial culture started from 5 × 10^5^ cells per well on 96-well plates. For inoculum control, 10 µL of untreated culture was resuspended in a 10 mL culture medium and spread (100 µL) on agar plates to enumerate c.f.u. after overnight incubation in 37 °C. The MIC of ITCs was defined at the lowest concentration inhibiting visible growth of bacteria after overnight incubation (20 h) at 37 °C and determined spectrophotometrically using EnSpire microplate reader (Perkin Elmer Singapore Pte. Ltd., Singapore). The MBC was measured by subculturing the cultures used for MIC determination onto fresh agar plates. MBC was the lowest concentration of a drug that results in killing 99.9% of the bacteria being tested.

### 3.3. Time-Kill Assay

The time-kill test was performed according to the CLSI guidelines and Zhou et al. [70]. *V. cholerae* O395N1 *toxT*::*lacZ* strain was grown in Mueller–Hinton medium on 96- well plates. After ITC (10× MIC, 5× MIC, 2× MIC, MIC) addition, samples were collected at 0, 2, 4, 6, 8, 10, 12, and 24 h of growth, and the colony-forming units were numbered by plating serial dilutions onto MH agar plated after incubation for 24 h at 37 °C in microaerobic conditions. The experiment was performed in triplicate.

### 3.4. Biofilm Eradication

Biofilm formation of *V. cholerae* O395N1 *toxT*::*lacZ* was assessed in sterile 96-well polystyrene microtiter plates. Overnight cultures were added to microtiter plates to form biofilms for 24 h. The plates were incubated for 24 h at 30 °C. Afterwards, the medium was removed, and the biofilms were exposed to SFN or PEITC (at MIC and 5 × MIC) for 1 h. Sterile Mueller–Hinton medium was used as a control. Then, the biofilm mass and viability were determined in crystal violet (1%) and MTT (0.5%) assays, respectively, according to the manufacturer’s protocols.

### 3.5. Measurement of DNA and RNA Synthesis

The estimation of nucleic acid synthesis was performed by measurement of incorporation of radioactive precursors, [^3^H] thymidine for DNA, and [^3^H] uridine for RNA, according to the procedure described in [22]. Briefly, overnight bacterial cultures were diluted in fresh LB medium 1:100 and grown to A_600_ of 0.1. Isotope was added at 150 μCI. At the time points, 50 µL of samples were withdrawn and placed onto Whatman filter paper and then transferred immediately to ice-cold 10% trichloroacetic acid (TCA), next Whatman filter paper was washed in 5% TCA and twice in 96% ethanol. The filters were dried, and radioactivity was measured in a scintillation counter MicroBeta2 (PerkinElmer, Downers Grove, IL, USA). Results from three independent experiments were normalized to bacterial culture density.

### 3.6. RNA Extraction, Reverse Transcription, Primers, and qPCR Analysis

The RT-qPCR analyzes of gene expression were performed as described elsewhere [71]. Briefly, *V. cholerae* (wild type) was cultivated aerobically at 37 °C overnight, then moved to a fresh medium (1:100). The culture was grown to OD_600_ = 0.1. and compounds were added as indicated. Samples were taken at time zero, 15, 45, and 60 min. Total RNA was extracted using RNeasy Mini Kit (Qiagen GmbH, Hilden, Germany) according to the manufacturer’s protocol. The total RNA concentration and purity of samples were assessed using Agilent RNA Kits (2100 Bioanalyzer System, Agilent Technologies, Santa Clara, CA, USA). Next, reverse transcription was performed using EvoScript Universal cDNA Master (Roche, Basel, Switzerland). Subsequently, quantification of transcripts in samples was performed by real-time qPCR analysis using SYBR Green-based method. The experiment was carried out in LightCycler^®^ 480 instrument (Roche Diagnostics AG, Rotkreuz, Switzerland). Briefly, the qPCR reaction (10 µL) contained: 1× SYBR Green I Master Mix, oligonucleotides (primer list, Appendix A), and 1 µL of cDNA template (20 ng/µL) obtained in a previous step. The qPCR assay was performed with the following amplification program: activation of the enzyme at 95 °C for 10 min., followed by 45 cycles of 95 °C for 10 s, 60 °C for 10 s, and 72 °C for 10 s. Melting curve analysis of the PCR products was performed to ensure the specificity of the products. Normalization was done by the amount of the total RNA in each RT reaction and by the level of expression of a reference gene *recA.*

### 3.7. In Vitro Cultures Toxicity Assay

The Hela and Vero cells were cultivated in DMEM (Gibco, Waltham, MA, USA) supplemented with 10% horse serum (Gibco, Waltham, MA, USA) along with penicillin G and streptomycin and maintained at 37°C in an atmosphere of humidified 5% CO_2−_ 95% air in a CO_2_ incubator. In order to evaluate the activity of CT in HeLa or Vero cells, bacterial lysates were prepared using WT strain as we reported previously [25] with minor changes. Briefly, overnight cultures were grown in LB medium (30 °C, 165 rpm). After 100-fold dilution in fresh LB medium, the cultivation was continued until the OD_600_ = 0.1 was reached. Then, SFN (½ MIC) and PEITC (¼ MIC) were added to the culture and incubated with shaking for 360 min. Next, lysates were withdrawn, samples normalized to 10^11^ cells per group and centrifuged (4200 rpm, 20 min, 4 °C), the pellet was suspended in 1 x PBS and subjected to a 3-fold freezing and thawing cycle in liquid nitrogen, the lysate was filtered using syringe filters (PVDF, 0.22 μm, Roth, Karlsruhe, Germany).

Cell viability was determined by the MTT method as previously described [25]. Cells were plated in four repetitions (2 × 10^3^ cells/well) of 96-well flat-bottomed plates and incubated overnight prior to ITCs and cholera toxin exposure. Then the medium was changed to a fresh one supplemented with test compounds. 5µL of diluted bacterial lysates (1 mL medium + 40 µL bacterial lysates) were added to 95 µL DMEM. The control experiments with appropriate concentrations of compounds were performed. After 48 h medium was changed to MTT solution (5 mg/mL in 1 × PBS), cells were incubated with MTT solutions for 2 h. Then the violet formazan product was dissolved in 100 µL DMSO and measured at 570 nm and 660 nm using Victor3 plate reader (Wallac Oy, Perkin Elmer, Turku, Finland).

### 3.8. Assessment of (p)ppGpp Accumulation Levels in Cells after ITC’s Treatment

The measurement of cellular guanosine tetra-phosphate (ppGpp) level was based on the [^32^P]-labeling of nucleotides method developed by thin-layer chromatography on PEI-cellulose plates [72]. Fresh *V. cholerae* culture was suspended in PBS to obtain OD_600_ = 1.0 on the McFarland scale as measured by a Densila-Meter II (ErbaLachema, Brno, Czech Republic). The suspension was diluted at a 1:10 ratio in a minimal MOPS medium containing 0.4 mM KH_2_PO_4_ (0.2% casamino acids). Then, the [^32^P]-orthophosphoric acid was added to the cultures to a final concentration of 150 μCi/mL, and bacteria were grown at 37 °C for 1 h with shaking 165 rpm/min. After this time, the bacterial cultures (150 μL) were transferred to wells in a 96-well plate. SFN, PEITC in a range of concentrations were added. The amino acid starvation was induced by adding an L-serine amino acid- serine hydroxamate (SHX) analogue to a final concentration of 0.8 mg/mL to the bacterial cultures. Samples were collected at the indicated time, then extracted with formic acid (13 M) in two cycles of freezing and thawing in liquid nitrogen. Samples were centrifuged (5000× *g*, 5 min), and the nucleotides present in the supernatant were separated by thin layer chromatography on a polyethyleneimine (PEI) TLC plate with cellulose in 1.5 M potassium phosphate buffer and analyzed with Typhoon 9200 (GE Healthcare, Sweden).

### 3.9. Galleria Mellonella Model of Infection Procedures

*G. mellonella* larvae TruLarve were used to perform analysis (BioSystems Technology, Credition, UK). The survival and cytotoxicity experiments were performed according to the manufacturer’s protocols. Larvae were injected with 10 µL of solution (bacterial suspension, antibiotic or PBS) into the last proleg using a blunt tip 50 µL syringe (Hamilton). Surface sterilization of insects with EtOH was performed prior to each loading. The larvae were then incubated at 37 °C, and survival and melanization were recorded in the next 96 h. Larvae were scored as dead when they ceased moving, changed from their normal pale cream coloration to brown, and failed to respond when gently manipulated with a pipette tip. PBS-injection controls were used. Survival 96 h post-infection was recorded.

#### 3.9.1. Virulence and Treatment of *V. cholerae* Infection in *G. mellonella*

The wax moths, whose weights were 250 ± 25 mg, with no symptoms of melanization were divided into different groups. Fist, the virulence of *V. cholerae* O395 was assessed by challenging larvae with range of c.f.u/mL (10^6^, 5 × 10^5^, 5 × 10^4^, 5 × 10^3^, 5 × 10^2^) of bacteria washed and suspended in PBS. The safety of SFN and AZT administration was tested at doses 25, 10, 5 mg/kg, and 20 mg/kg, respectively. Next, the effectiveness of drugs was assessed by pretreatment of larvae by injection (10 µL) of SFN (25 mg/kg), AZT (20 mg/kg), or both agents in mixture into the last left proleg at 3 h before inoculum (5 × 10^2^ c.f.u) administration (10 µL) into the right proleg. The survival rate was then monitored at 24 h intervals. Fifteen larvae per group were used and each experiment was assessed in triplicate.

#### 3.9.2. FITC Labeling of Bacteria

FITC labeling of bacteria was performed as described elsewhere [73]. Briefly, bacteria were harvested from exponential phase cultures and heat-killed (1 h at 70 °C, 10^9^/mL). Then, cells were incubated in 0.5% carbonate buffer (pH 9.5) containing FITC (0.1 mg/mL) for 30 min at 37 °C. The FITC-conjugated bacteria were then washed three times in PBS and kept at −20 °C until use.

#### 3.9.3. In Vivo Phagocytosis Assay

For in vivo assays, 10 μL of PBS solution containing approximately 5 × 10^7^ cells of FITC-labeled *V. cholerae* were injected into *G. mellonella* via the last left proleg using a 50 μL Hamilton syringe. After 1 and 2 h at 28 °C, the insects were surface sterilized and bled directly onto a glass coverslip in a drop of PBS. After 10 min of incubation at room temperature, a fluorescence-quenching assay was achieved by adding 0.5 mL of trypan blue dye (0.5% in PBS) and incubated for 15 min. Cells were then washed in NaCl solution (8 g/L) and fixed in 3.7% paraformaldehyde in PBS for 10 min. Phagocytosis was assessed by fluorescence microscopy using Leica DMI4000B microscope fitted with a DFC365FX camera (Leica Microsystems, Germany). To quantify the rate of phagocytosis and the phagocytic index for each coverslip, five fields that contained at least 100 hemocytes were examined. The rate of phagocytosis was calculated according to the following formula: Phagocytic rate = ((phagocytic hemocyte)/(total hemocyte)) × 100%.

### 3.10. Transmission Electron Microscopy (TEM)

The MIC of SFN and PEITC were added to bacterial cultures (OD_600_ = 0.4) and incubated in 37 ◦C. Nontreated cultures were used as control. Next, at the time indicated, bacteria were pelleted by centrifugation (4000× *g*, 10 min) and resuspended in a fresh medium. 5 μL of the diluted (100 times) bacteria culture (before or after laser treatment) was dropped and dried on a copper grid for TEM observation. Electron microscopic analyses were performed using the Tecnai Spirit BioTWIN microscope (FEI Company, Eindhoven, Netherlands) at 120 kV.

### 3.11. Statistical Analysis

The data are expressed as the means ± SD. Two-tailed unpaired Student’s *t*-tests were used to analyze the differences between experimental groups with the exception of Kaplan–Meier survival assays of *G. mellonella,* where Chi-square analyzes were performed. All experiments were repeated at least three times independently for reproducibility. Technical repeats were performed where indicated. *p* values < 0.05 were considered statistically significant. GraphPad Prism 8 software (GraphPad Software, San Diego, CA, USA) was used for statistical analysis.

## 4. Conclusions

Our results indicate a two-way mechanism of action of ITC against *V. cholerae*: inhibition of bacterial growth (both in plankton and biofilm) and suppression of toxin production. They efficiently protect mammalian cells and *G. mellonella* larvae against *V. cholera* toxicity at concentrations safe for the host cells. Therefore, they constitute promising candidates for the treatment of *V. cholera* infections, either as monotherapy or as a combination with antibiotics. Stringent response activation by ITC is considered the most likely mechanism underlying metabolic disturbances in treated bacteria. However, further studies are needed to elucidate this phenomenon and its implication for future utility in the control of *V. cholerae* infections.

## Figures and Tables

**Figure 1 ijms-22-10187-f001:**
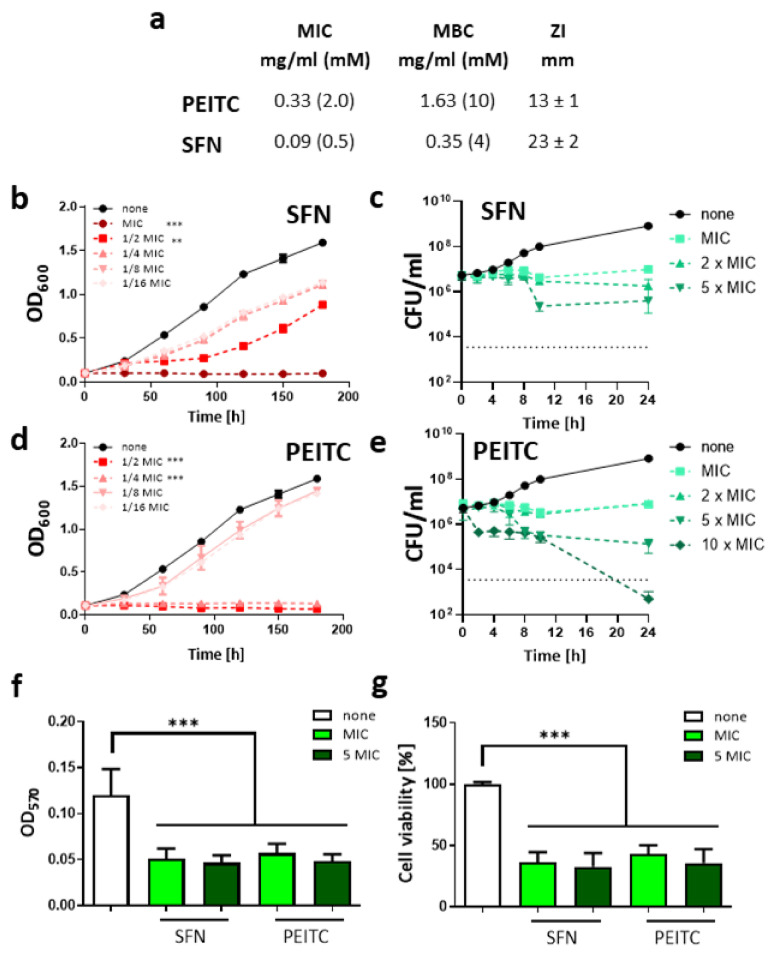
Antimicrobial activity of PEITC and SFN against *V. cholerae.* The *V. cholerae* susceptibility tests evaluated as minimal inhibitory (MIC) and bactericidal (MBC) concentration and zone inhibition assays. (**a**) The growth inhibition of *V. cholerae* treated with various concentrations of SFN (**b**) and PEITC (**d**). The killing curves of bacteria treated with SFN (**c**) and PEITC (**e**) in doses equal or greater than MIC. The results presented are mean values from 3 independent experiments with SD indicated. The reduction/inactivation of formed biofilm was assessed by crystal violet staining (**f**) and MTT assay (**g**). The significance of differences between treated samples and the controls was tested using the Student’s *t*-test. Statistical significance is marked with asterisks: *** *p* < 0.001; ** *p* < 0.01.

**Figure 2 ijms-22-10187-f002:**
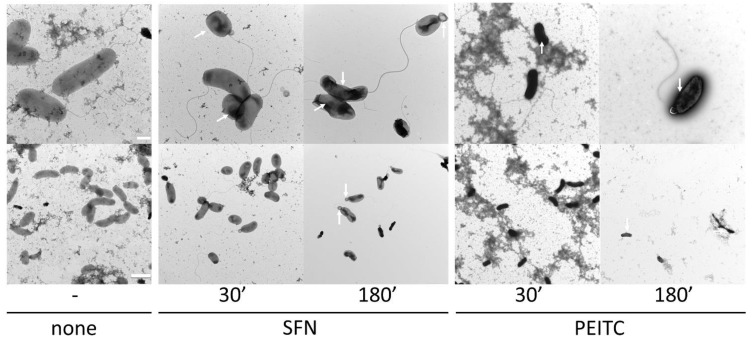
TEM analysis of morphological changes after ITC treatment. Cells treated with MIC of SFN or PEITC after different treatment times (30, 180 min) at 37 °C, or none for control experiment. The white arrows indicate shape malformations and cytosol leakage. White bars indicate the scale: 0.5 µm and 2 µm for the upper panel and the bottom panel, respectively.

**Figure 3 ijms-22-10187-f003:**
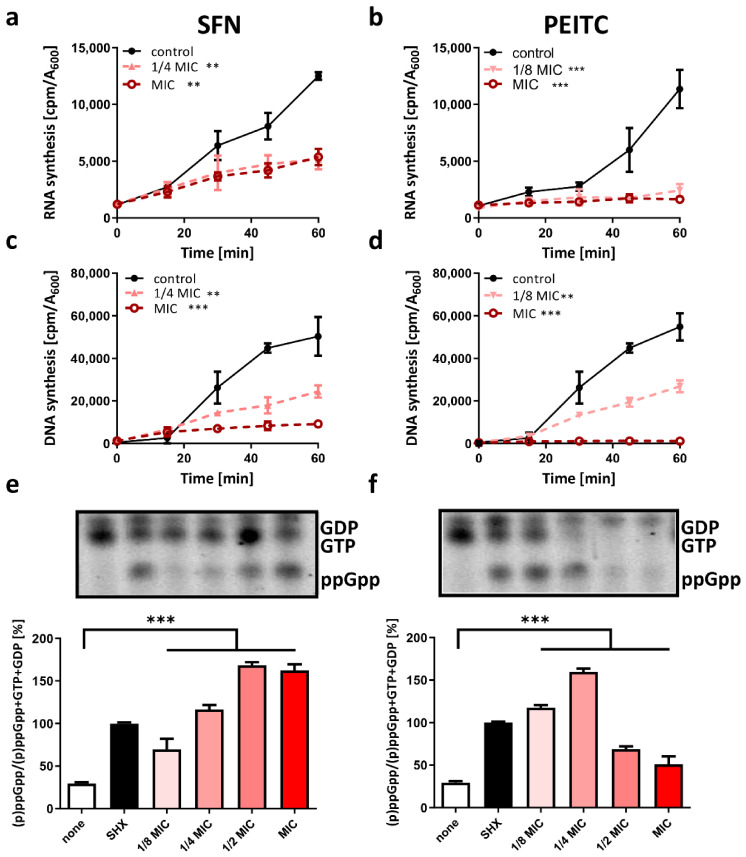
SFN and PEITC impair nucleic acid synthesis and activate stringent response and (p)ppGpp alarmone accumulation in *V. cholerae*. The effect of SFN (**a**) and PEITC (**b**) on RNA synthesis was assessed in [^3^H]uridine incorporation assay. The effect of SFN (**c**) and PEITC (**d**) on DNA synthesis assessed in [^3^H]thymidine incorporation assay. The (p)ppGpp alarmone level in cells treated (15 min) with SFN (**e**) and PEITC (**f**). Bacteria were grown in MOPS labeling medium in the presence of 150 µCi/mL [^32^P]orthophosphoric acid with various concentrations of ITC or starved for amino acids by the addition of SHX. The labeled guanosine nucleotides separated on TLC are presented in the panels. The percentage of ppGpp in the total pool of guanosine nucleotides in various conditions is presented on the graphs. The results presented are mean values from 3 independent experiments. The significance of differences between the results and the control was tested using the Student’s *t*-test. Statistical significance is marked with asterisks: *** *p* < 0.001; ** *p* < 0.01.

**Figure 4 ijms-22-10187-f004:**
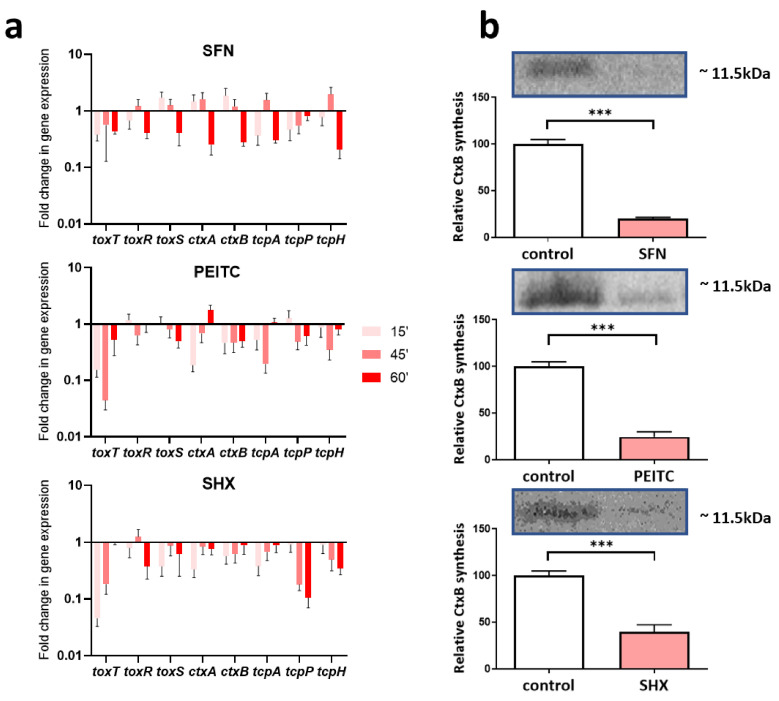
The expression of *V. cholerae* virulence factors is decreased by ITC treatment. (**a**) The genes’ expression was assessed using RT-qPCR method. Transcriptional levels of various virulence-related genes were analyzed by qRT-PCR in the stringent response inducing conditions, namely treatment with SFN (¼ MIC), PEITC (1/8 MIC), SHX (0.8 mg/mL) for 15, 45, and 60 min at 37 °C. The relative transcription level of each gene was compared to the *recA* gene used as an internal control. ΔΔCt was calculated as: ΔCt (test)−ΔCt (calibrator). Ratio= efficiency^−ΔΔCt^. (**b**) CT protein was detected by Western blot using anti-CTB rabbit monoclonal antibodies. Samples were collected after 20 h. Toxin production was quantified densitometrically by the Quantity One (Bio-Rad). The results presented are mean values from 3 independent experiments. The Student’s *t*-test statistical significance is marked with asterisks, *** *p* < 0.001.

**Figure 5 ijms-22-10187-f005:**
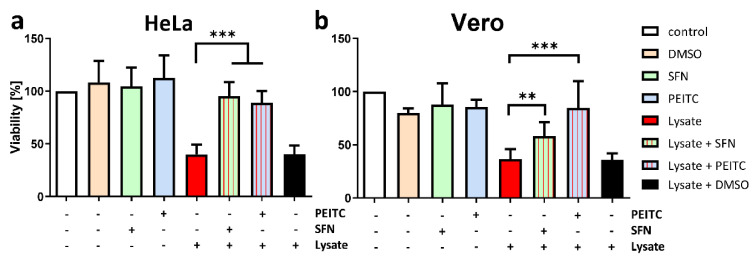
SFN and PEITC reduce the toxicity of *V. cholerae* against HeLa and Vero cell cultures. Bacterial lysates were obtained after centrifugation of 10^11^ cells from culture supplemented with ITC and culture without treatment (negative control). HeLa and Vero cells (2 × 10^3^) were grown in DMEM medium. The viability of HeLa (**a**) and Vero (**b**) cells after treatment with ITC and bacterial lysate was measured using the MTT assay. Cytotoxicity control of the vehicle (DMSO), SFN, and PEITC were assessed as indicated on the graphs. The results presented are mean values from 3 independent experiments. The significance of differences between the results and the positive control (bacterial lysate with CT) was tested using the Student’s *t*-test. Statistical significance is marked with asterisks: *** *p* < 0.001; ** *p* < 0.01.

**Figure 6 ijms-22-10187-f006:**
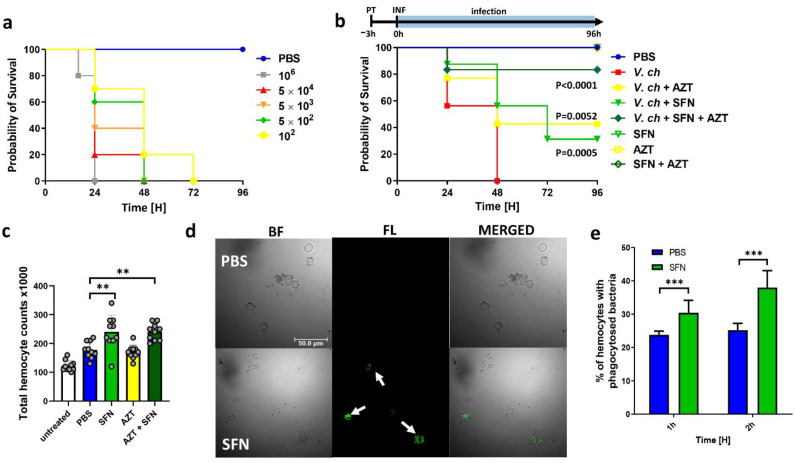
*G. mellonella* larvae pretreatment with SFN improve their survival rate after *V. cholerae* infection. (**a**) Kaplan–Meier survival analysis of *Galleria mellonella* larvae infected with *Vibrio cholerae* O395. Larvae (n = 15) were injected with (10 µL) PBS (blue circle) or *V. cholerae* suspension in PBS at range of c.f.u/per dose: 10^2^ (yellow circle) or 5 × 10^2^ (green deltoid) or 5 × 10^3^ (orange triangle) or 5 × 10^4^ (red triangle) or 10^6^ c.f.u (gray squares). (**b**) Treatment efficacy assessed by survival analysis. Larvae (n = 15) were pre-treated (3 h before infection) with SFN (light green triangle) or AZT (yellow cross) or SFN + AZT (dark green deltoid). Infection groups (filled symbols) were then injected (t = 0 h) with 5 × 10^2^ c.f.u of bacterial suspension (*V. ch*), not treated control is represented by red squares. Not infected control groups are marked as empty symbols. Double injection with PBS was performed in the uninfected control group (blue circle). Larvae were injected and monitored as indicated on the timeline above graph. PT- pretreatment INF- infection. (**c**) Total hemocyte infiltration to hemolymph (3 h) in larvae injected with PBS (blue), SFN (light green), AZT (yellow), SFN + AZT (dark green) or not treated (white), individual. (**d**) Representative images of infected hemocytes are shown. The fluorescent signal of FITC-labeled bacteria is indicated by arrows. Scale bar upper left panel denotes 50 μm and applies to the rest of the micrographs. BF-bright light; FL-fluorescence (**e**) Quantification of the phagocytosis percentage. The significance of differences between the results and the control was tested using the Student’s *t*-test. Statistical significance is marked with asterisks: *** *p* < 0.001; ** *p* < 0.01.

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
