# Peer review of "Dietary Isothiocyanates, Sulforaphane and 2-Phenethyl Isothiocyanate, Effectively Impair Vibrio cholerae Virulence"

_ijms, 2021, doi:10.3390/ijms221910187_

Round 1

Reviewer 1 Report

Revision of manuscript ijms-1339802

Dear Authors,

Your manuscript entitled “Dietary isothiocyanates, sulforaphane and 2-phenethyl isothiocyanate, effectively impair Vibrio cholerae virulence” is a very interesting study on the effects of 2 plants metabolites on V. cholerae. Different aspects were evaluated (antimicrobial activity, biofilm prevention, biofilm “destruction”, toxins genes expression, effect on cell lines and insects). The work is complete, well planned and conducted. Results are interesting, well presented and discussed.

I did not find inaccuracies or problems. A very good work.

Sincerely

The Reviewer

Author Response

Dear Reviewer,

Thank you very much for the very positive opinion on our manuscript.

Reviewer 2 Report

Thank you for the opportunity to review this manuscript. This manuscript reported the anti-Vibrio cholerae activities of plant-derived compounds such as isothiocyanates and phytoncides. The authors show that sulforaphane and 2-phenethyl isothiocyanate have the ability to inhibit V. cholerae growth both in vitro and in vivo. This manuscript is well written, and the information here might be helpful for readers. However, the authors need to address some concerns before publication.  Please see my specific comment stated below.

Major Comments

  1. The major concern of this manuscript is the lack of control experiments (both positive controls and negative controls) is missing. For instance, the positive controls for MIC and MBC in Fig.1, the positive controls in Fig.2, and both positive and negative controls for Fig.5,
  2. Why is the effect of SHX missing in Fig.4b?
  3. Please discuss the reason why the effects of SFN and PEITC are different among these cell lines in Fig.5.
  4. The data about in vivo toxicity is mandatory in Fig.6, e.g., body weight loss of animals received with SFN and AZT.

Minor comments:

  1. Line 14: Vibrio cholerae should be spelled out.
  2. Line 20: MIC and MBC should be spelled out.
  3. Line 32: Vibrio cholerae should be spelled out.
  4. Lines 86-87: These pathogens should be spelled out.
  5. Line 112: MIC should be spelled out.
  6. Line 118: Please delete “tested” or “assessed”.
  7. Line 119: OD should be spelled out.

Author Response

We would like to thank the Reviewer for the constructive opinion and for bringing up the issues which helped us to improve the manuscript. Please find below point-by point responses to the specific questions and concerns.

Major Comments

  1. The major concern of this manuscript is the lack of control experiments (both positive controls and negative controls) is missing. For instance, the positive controls for MIC and MBC in Fig.1, the positive controls in Fig.2, and both positive and negative controls for Fig.5,

Response

The positive control – chloramphenicol - for MIC and MBC are now included in the text (lines 118-120, 122-124).

In the Figure 2, the changes in the cell morphology upon ITC treatment are depicted where the negative control is lack of any compound added, and the effects can be compared to normally looking healthy bacterial cells. The cell shrinkage and the deformation of the shape as well as a condensation of the cytoplasm are indicated by arrows in the Figure 2. In the Figure 5 the negative controls consist of not treated cells (first columns in both panels) and cells treated with DMSO (columns added to both figures), and they are followed by the combination of lysate treated or not treated with specific ITC, illustrating the effects of the treatment.

  1. Why is the effect of SHX missing in Fig.4b?

Response

The SHX panel is now included in the Figure 4 and mentioned in the text (lines 334-335)

  1. Please discuss the reason why the effects of SFN and PEITC are different among these cell lines in Fig.5.

Response:

The possible reason for observed differences in the effects of ITC on various cell lines is now discussed in the text (lines 358-361)

  1. The data about in vivo toxicity is mandatory in Fig.6, e.g., body weight loss of animals received with SFN and AZT.

Response:

The results of the in vivo toxicity of the compounds used in the experiments are now included in the Fig. S2

For the Galleria mellonella model, the standard procedure to assess the bacterial/compound effect on the larvae is observation of insect vitality and condition (e.g. mobility, melanisation) and survival (as described in Bokhari et al, 2017 and Nuidate et al, 2016), not a body weight (which would be very difficult to assess) so we adhered to these protocols.

Minor comments:

  1. Line 14: Vibrio cholerae should be spelled out.
  2. Line 20: MIC and MBC should be spelled out.
  3. Line 32: Vibrio cholerae should be spelled out.
  4. Lines 86-87: These pathogens should be spelled out.
  5. Line 112: MIC should be spelled out.
  6. Line 118: Please delete “tested” or “assessed”.
  7. Line 119: OD should be spelled out.

Response:

The requested changes have been made in the text in the line 20-21, 116, 124 and 125. The requested spelling out of the pathogen names is already present in the text.

Reviewer 3 Report

The authors approach an utmost important subject under the worldwide extension of the antibiotic resistance phenomenon, that of alternative therapies in a very severe pathology, induced by V. cholerae. Plant extracts were intensively studied in later years as alternative sources for antibiotics, due to their higher bioavailability, lesser side effects and accessibility, based on ethno-pharmaceutical practices.

In this research, the authors have an extremely complex, multifaceted approach to analysing the potential preventive/therapeutic effects of Sulforaphane (SFN) and 2-phenethyl isothiocyanate (PEITC) in vibriosis.

The study would very much benefit of a more organised and explicit presentation of the methodology, some issues concerning the methods being mentioned in the Results and discussion, while some of the relevant result, enhancing the understanding of the whole research, being in the supplementary material (Galleria mellonella surrogate host of V. cholerae infection- graph). 

There is a need for a more explicit English, on some occasions the expressions used being obsolete, difficult to understand. A more organised and explicit presentation of the results and discussion would definitely add  value to the paper for its readers.

Review of Dietary isothiocyanates, sulforaphane and 2-phenethyl isothio- 2 cyanate, effectively impair Vibrio cholerae virulence 3 Klaudyna Krause1 , Agnieszka Pyrczak-Felczykowska3 , Monika Karczewska1 , Magdalena Narajczyk4 , Anna Her- 4 man-Antosiewicz2 , Agnieszka Szalewska Pałasz1, Dariusz Nowicki1*

International Journal of Molecular Science ID 1339802

Line 27 “might be considered as a promising antibacterial agents in V. cholerae infections.” Replace with “might be considered promising antibacterial agents in V. cholerae infections.”

Lines 39-40 Please rephrase  “the treatment capacity of infected patients which has created a major 39 challenge to the development of an efficient therapy”, treatment capacity of infected patients sounds somewhat ambiguous

Line 57-58 “V. cholerae strains are commonly present in their natural reservoirs: the bacterial flora 57 of both freshwater and marine environments [10]” – the natural reservoirs are aquatic environments not the bacterial flora

Line 162 “Just relatively high concentration of PEITC (10 x MIC) showed the bactericidal effect, but  it could be difficult to consider it for the use in vivo due to its cytotoxic effect.”  Could the authors estimate the advantage of using the ITC in therapy in case their proven effect against Vibrio is just bacteriostatic?

Line 184-187 “These observations indicate that ITC can be used not only to inhibit bacterial exponential growth but importantly, also diminish their biofilm formation and eradicate already formed biofilms, which highlights important preventive role of ITC against pathologies induced by V. cholerae.”  The authors should also discuss the clinical preventive potential based on the obtained in vitro results (if any forseen)

Lines 358- 362 include Materials and methods

“We first established the bacterial load appropriate for survival experiments (Figure S3). The 5 x 102 bacteria per larvae were chosen for further analyzes as they resulted in 50% death rate at 24 h after infection. SFN safety (at the 25, 10, 5 360 mg/kg doses) was assessed and the tested concentrations were found to be non-toxic and no symptoms of melanisation, associated with the worsening of the insect condition, were  observed after the treatment (data not shown).”

Lines 364- 365 this procedures are not clear from materials and methods

 “We demonstrate that pre-treatment of  larvae with 25 mg/kg SFN administrated 3h before infection improved the survival of G. mellonella”

“Additionally we analyzed potency of SFN to enhance azithromycin (20 mg/kg) activity against  V. cholerae . …The observed larvae survival rate after single dose of azithromycin treatment was surprisingly similar to the level observed in the objects which received SFN alone. Interestingly, administration of two compounds together decreased mortality in V. cholerae challenged larvae to 20 % after 96 h as compared with infected control.”

Line 525 “3.9. Galleria mellonella model of infection procedures” could more clearly explained”

Line  528 “Larvae were injected (10 µl) into last proleg using blunt tip 50 µl syringe (Hamilton). “ with 10 µl  of what?

Line 537 “FITC labeling of bacteria was performed by the method [69].” – please mention the method used, not only the reference number in brackets

Line 565 “Unpaired Student’s t tests two-tailed were used to analyze” please rephrase in proper English

Author Response

We thank the Reviewer for the detailed review of our manuscript which helped in organizing the our work in more comprehensive way. The text has been corrected by the professional language editor. The detailed changes in the manuscript are described as a responses to points in the review.

Review of Dietary isothiocyanates, sulforaphane and 2-phenethyl isothio- 2 cyanate, effectively impair Vibrio cholerae virulence 3 Klaudyna Krause1 , Agnieszka Pyrczak-Felczykowska3 , Monika Karczewska1 , Magdalena Narajczyk4 , Anna Her- 4 man-Antosiewicz2 , Agnieszka Szalewska PaÅ‚asz1, Dariusz Nowicki1*

International Journal of Molecular Science ID 1339802

Line 27 “might be considered as a promising antibacterial agents in V. cholerae infections.” Replace with “might be considered promising antibacterial agents in V. cholerae infections.”

Response:

The requested change has been made (line 28).

Lines 39-40 Please rephrase  “the treatment capacity of infected patients which has created a major 39 challenge to the development of an efficient therapy”, treatment capacity of infected patients sounds somewhat ambiguous

Response:

The sentence has been rephrased.

Line 57-58 “V. cholerae strains are commonly present in their natural reservoirs: the bacterial flora 57 of both freshwater and marine environments [10]” – the natural reservoirs are aquatic environments not the bacterial flora

Response:

This sentence has been reformulated (lines 59-60).

Line 162 “Just relatively high concentration of PEITC (10 x MIC) showed the bactericidal effect, but  it could be difficult to consider it for the use in vivo due to its cytotoxic effect.”  Could the authors estimate the advantage of using the ITC in therapy in case their proven effect against Vibrio is just bacteriostatic?

Response:

Quite a number of antibiotics used in the routine anti-infection therapy act as bacteriostatic agents thus the usage of ITC to combat cholera infection can be also proposed. It is discussed now in the manuscript (lines 170-172).

Line 184-187 “These observations indicate that ITC can be used not only to inhibit bacterial exponential growth but importantly, also diminish their biofilm formation and eradicate already formed biofilms, which highlights important preventive role of ITC against pathologies induced by V. cholerae.”  The authors should also discuss the clinical preventive potential based on the obtained in vitro results (if any forseen)

Response:

The possibility of using ITC as a preventive agent in the food industry has been already proposed, so we refer now to this and hypothesize on plausible usage of SFN and PEITC to prevent e.g. seafood related cholera infection (lines 197-202)

Lines 358- 362 include Materials and methods

“We first established the bacterial load appropriate for survival experiments (Figure S3). The 5 x 102 bacteria per larvae were chosen for further analyzes as they resulted in 50% death rate at 24 h after infection. SFN safety (at the 25, 10, 5 360 mg/kg doses) was assessed and the tested concentrations were found to be non-toxic and no symptoms of melanisation, associated with the worsening of the insect condition, were  observed after the treatment (data not shown).”

Lines 364- 365 this procedures are not clear from materials and methods

 “We demonstrate that pre-treatment of  larvae with 25 mg/kg SFN administrated 3h before infection improved the survival of G. mellonella”

“Additionally we analyzed potency of SFN to enhance azithromycin (20 mg/kg) activity against  V. cholerae . …The observed larvae survival rate after single dose of azithromycin treatment was surprisingly similar to the level observed in the objects which received SFN alone. Interestingly, administration of two compounds together decreased mortality in V. cholerae challenged larvae to 20 % after 96 h as compared with infected control.”

Line 525 “3.9. Galleria mellonella model of infection procedures” could more clearly explained”

Response:

We agree with the Reviewer that the procedures involving Galleria mellonella model have not been described sufficiently. Thus, now the extended and detailed procedures are included in the Material and methods sections (3.9.1., lines 580-589). The timeline of the experiment is included in the Fig 6B, the bacterial load assessment is now presented in the Fig. 6A (instead of supplementary data), and the potential toxicity and safety of the used compounds is presented on the Fig 2S.

Line  528 “Larvae were injected (10 µl) into last proleg using blunt tip 50 µl syringe (Hamilton). “ with 10 µl  of what?

Response:

Any used compound or PBS as a control has been injected in this volume. The clarification on this is now included in the text (lines 571-572)

Line 537 “FITC labeling of bacteria was performed by the method [69].” – please mention the method used, not only the reference number in brackets

Response:

The relevant reformulation has been made in the text (lines 592-594).

Line 565 “Unpaired Student’s t tests two-tailed were used to analyze” please rephrase in proper English

Response:

The sentence has been rephrased as requested (line 622).

Round 2

Reviewer 2 Report

I have gone through the revised manuscript, and I think all my comments have been adequately addressed. Thank you again for the opportunity to review this manuscript.

Reviewer 3 Report

The authors followed the suggestions of the reviewer upgrading the quality of the paper